# Therapeutic Potential of AAV1-Rheb(S16H) Transduction against Neurodegenerative Diseases

**DOI:** 10.3390/ijms22063064

**Published:** 2021-03-17

**Authors:** Youngpyo Nam, Gyeong Joon Moon, Sang Ryong Kim

**Affiliations:** 1Brain Science and Engineering Institute, Kyungpook National University, Daegu 41944, Korea; blackpyo2@naver.com; 2Center for Cell Therapy, Asan Institute for Life Science, Asan Medical Center, Seoul 05505, Korea; gj.moon@amc.seoul.kr; 3Department of Convergence Medicine, University of Ulsan College of Medicine, Seoul 05505, Korea; 4School of Life Sciences, Kyungpook National University, Daegu 41566, Korea; 5BK21 FOUR KNU Creative BioResearch Group, Kyungpook National University, Daegu 41566, Korea

**Keywords:** neurotrophic factor, Rheb(S16H), neurodegenerative disease, Alzheimer’s disease, Parkinson’s disease

## Abstract

Neurotrophic factors (NTFs) are essential for cell growth, survival, synaptic plasticity, and maintenance of specific neuronal population in the central nervous system. Multiple studies have demonstrated that alterations in the levels and activities of NTFs are related to the pathology and symptoms of neurodegenerative disorders, such as Parkinson’s disease (PD), Alzheimer’s disease (AD), and Huntington’s disease. Hence, the key molecule that can regulate the expression of NTFs is an important target for gene therapy coupling adeno-associated virus vector (AAV) gene. We have previously reported that the Ras homolog protein enriched in brain (Rheb)–mammalian target of rapamycin complex 1 (mTORC1) axis plays a vital role in preventing neuronal death in the brain of AD and PD patients. AAV transduction using a constitutively active form of Rheb exerts a neuroprotective effect through the upregulation of NTFs, thereby promoting the neurotrophic interaction between astrocytes and neurons in AD conditions. These findings suggest the role of Rheb as an important regulator of the regulatory system of NTFs to treat neurodegenerative diseases. In this review, we present an overview of the role of Rheb in neurodegenerative diseases and summarize the therapeutic potential of AAV serotype 1 (AAV1)-Rheb(S16H) transduction in the treatment of neurodegenerative disorders, focusing on diseases, such as AD and PD.

## 1. Introduction

Neurodegenerative diseases are debilitating disorders characterized by the gradual loss of anatomically or physiologically related function or structure of the central nervous system and include various diseases with different pathological patterns and clinical manifestations, such as Alzheimer’s disease (AD), Huntington’s disease (HD), Parkinson’s disease (PD), and amyotrophic lateral sclerosis (ALS) [1,2,3]. These diseases are pathophysiologically diverse, causing cognitive and memory impairments or affecting a person’s ability to move, breathe, and speak [4,5,6,7]. Although several studies have focused on the treatment of neurodegenerative diseases, a relevant treatment is yet to be developed. An in-depth understanding of the causes and mechanisms of each disease is essential for effective treatment of neurodegenerative diseases. These diseases are characterized by the selective loss of vulnerable populations of specific neurons due to various factors, such as increased number of reactive oxygen species [8], excitotoxicity [9], synaptic dysfunction [10], inflammation [11], impaired protein degradation systems [12], endoplasmic reticulum stress [13], and mitochondrial dysfunction [14]. Although the precise mechanism responsible for neuronal loss and functional disruption in these diseases still remains unknown, several studies have attempted treatment strategies through the protection of damaged neuronal cells. Several studies have also demonstrated that a decrease in the levels of neurotrophic factors (NTFs), such as brain-derived neurotrophic factor (BDNF) and ciliary neurotrophic factor (CNTF), is associated with the pathology of neurodegenerative diseases [15,16,17,18,19,20] and that this decrease is closely linked to neuronal cell death [21,22]. These findings suggest that sustained expression of NTFs is effective in protecting neurons in neurodegenerative diseases, such as AD and PD, and thus, the discovery and regulation of key molecules capable of regulating the production of NTFs is a potential therapeutic strategy for neurodegenerative diseases.

Ras homolog protein enriched in brain (Rheb) is a member of the Ras superfamily, which consists of Rheb1 and Rheb2 [23]. Rheb1 and Rheb2 proteins show a 51% similarity in their amino acid composition [24]. These proteins are expressed in high concentrations in various regions of the brain, including the hippocampus, cerebral cortex, frontal lobe, temporal lobe, and occipital pole [25,26]. The expression levels of Rheb are increased as an immediate early response to toxic stimulation, such as seizure and high-frequency-induced synaptic stimuli in an N-methyl-d-aspartate (NMDA)-dependent manner [26,27]. Further, Rheb can be activated by growth factors, such as nerve growth factor (NGF), epithelial growth factor, and fibroblast growth factor, in cultured neuronal cells [26], and it is associated with neuronal differentiation, growth, axonal regeneration, autophagy, energy homeostasis, amino acid uptake, and aging [24,28,29,30]. Previous studies have demonstrated that inhibition of the mammalian target of rapamycin (mTOR) accelerates autophagy, leading to the reduction of the levels of neurotoxic aggregate proteins such as Tau, α-synuclein, and ataxin [31,32,33], which are associated with diseases such as AD, PD, and spinal ataxia [34,35,36]. However, a recent study showed that Rheb activity was reduced in striatal tissues of patients with HD and mice with HD, and that activation of mammalian target of rapamycin complex 1 (mTORC1) by introducing caRheb, a constitutively active form, could mitigate the metabolic and degenerative phenotypes in striatal tissues of mice with HD [37]. Furthermore, there is accumulated evidence indicating the neuroprotective effects through increased Rheb expression for neurodegenerative diseases, including spinal cord injury, AD, and PD. It has been reported that the expression of adeno-associated virus vector (AAV)-caRheb in spinal neurons activates proteins downstream of mTOR and enhances intrinsic growth potential to cause axonal regeneration after spinal cord injury [28]. Another study showed that overexpression of Rheb depletes the levels of β-site amyloid precursor protein-cleaving enzyme 1 protein through proteasomal and lysosomal pathways, leading to reduction of Aβ generation [38]. Recently, it has been reported that activation of Rheb can protect neurons against neurotoxic conditions in the adult brain through neurotrophic interactions (such as BDNF, GDNF, and CNTF production) between neurons and astrocytes [39,40,41,42].

These findings suggest that Rheb involvement in neurodegenerative diseases and its important role in the production of various NTFs can be considered as one of the potential therapeutic targets for neurodegenerative diseases. This review provides insights into the involvement of the Rheb/mTORC1 signaling pathway in neurodegenerative diseases and the neuroprotective effects of Rheb as a potential therapeutic target, focusing on AD and PD.

## 2. Importance of Supporting Neurotrophic Factors as a Therapeutic Strategy for Neurodegenerative Diseases

Despite the several in-depth studies that have been conducted over previous decades on neurodegenerative diseases, these disorders remain poorly understood. Most neurodegenerative diseases, such as AD [43,44,45,46,47], PD [48,49,50,51,52], and HD [53,54,55,56,57] are accompanied by the loss of specific cell populations (Table 1), leading to functional and structural dysfunctions. Studies conducted in recent years have reported that the reduction of NTF levels and the specific requirements for neurotrophic support are among the factors that have been implicated in neuronal degeneration in these diseases. NTFs support cell survival, growth, differentiation, and synaptic plasticity of developing and mature neurons [58,59]. Many previous studies have demonstrated that expression-level changes in specific NTFs, such as BDNF, CNTF, and glial cell line-derived neurotrophic factor (GDNF), are associated with the pathogenesis of neurodegenerative diseases, such as AD [19,60,61], PD [62,63,64], HD [65,66], and ALS [67,68] (Table 2).

AD is a progressive psychiatric and neurodegenerative disease of unknown etiology. The clinical features of AD are a progressive decline in cognitive and memory processes and often difficulties with language [79,80,81]. The neuropathology of AD reveals extracellular deposits of amyloid plaques and the presence of numerous neurofibrillary tangles [82,83]. Neuronal degeneration and cognitive decline in patients with AD correlate with pathological changes in cholinergic neurons in several different regions [47,84,85]. The links between cholinergic changes and changes in NTF levels in AD have been emphasized by numerous studies. Several reports have indicated that the mRNA and protein expression levels of BDNF and tropomyosin receptor kinase B (TrkB) are reduced in the hippocampus and neocortex of postmortem brains of patients with AD [17,19,86]. Moreover, a recent study demonstrated that BDNF deficiency activates the JAK2/STAT3 pathway and leads to impairment in cognitive skill and synaptic plasticity [87]. Furthermore, the authors of that study confirmed that BDNF-deprivation-elicited events, including decreased BDNF expression and upregulated JAK2/STAT3 activation, also occur in the human AD brain. This observation supports that sustained BDNF expression and neurotrophic pathways are important factors in the protection of damaged neurons and consequently the inhibition of cognitive and memory decline. Previous preclinical and clinical studies have reported that GDNF, and its receptor alpha1 (GFRα1), levels in the brain were diminished in patients with AD and in a 3XTg AD mouse model [61,88,89]. In addition, it was reported that rearranged during transfection (RET) tyrosine kinase, which acts as a co-receptor of GDNF alongside GFRa1, was decreased in hippocampal neurons in a chronic cerebral hypoperfusion model of dementia [90]; moreover, overexpression of RET in AD neurons was related to neuronal survival [61,91]. Another in vitro and in vivo study showed that CNTF played a vital role in cognitive function recovery by stabilizing the levels of the synaptic protein PSD95 and synaptophysin in the Tg2576 AD mouse model [22]. These data suggest that reduced NTF levels contribute to neuronal degeneration in AD and that the regulation of NTFs is a potential treatment strategy for AD.

PD is a neurodegenerative disorder characterized by resting tremors, loss of postural reflexes, muscle rigidity, and bradykinesia, accompanied by festinating gait, cognitive decline, and postural malformations [92,93]. PD is pathologically characterized by the progressive death of heterogeneous populations of neurons, including dopaminergic, cholinergic, noradrenergic, and serotonergic neurons [48,49,50,51,52,94,95]. However, dopaminergic degeneration is considered a major cause of motor dysfunction and is the main target of existing and actively developed therapeutics. The degeneration of neurons in PD is due to various factors, including mitochondrial dysfunction, oxidative stress, excitotoxicity, and immune response [92,96]; however, the reduction of NTF levels in the degenerative region of the brain of patients with PD is also recognized as an important factor [97,98]. In postmortem examinations of the brains of controls and patients with PD, in situ hybridization study showed that BDNF mRNA is expressed by dopaminergic neurons in the substantia nigra pars compacta (SNpc) of healthy controls without known neurological disease, but the mRNA expression of SNpc BDNF was reduced in the brain of patients with PD. However, surviving dopaminergic neurons in the SNpc region of PD brain also expressed BDNF mRNA. This result indicates the importance of BDNF expression in the survival of SNpc neurons [64]. Other studies have measured the amount of BDNF expression in the blood of healthy controls and patients with PD and found a decreased amount in patients with PD, which correlated with motor impairment in these patients [99,100,101]. The therapeutic effect of GDNF is controversial because conflicting results have been reported regarding the protection and maintenance of neurons in PD. Absence of GDNF in parvalbumin-expressing interneurons using Cre recombinase-based mice modulated significant loss of catecholamine neurons in the nigrostriatal pathway and locus coeruleus [102]. In addition, inhibition of GDNF expression in GDNF-null mice showed progressive hypokinesia and selective reduction of tyrosine hydroxylase mRNA levels, accompanied by distinct catecholaminergic apoptosis in the substantia nigra and ventral tegmental area [103]. These data clearly show that GDNF is indispensable for the survival of adult catecholaminergic neurons under physiological conditions. The expression of GDNF was reduced in the brain of an animal PD model, and the delivery of GDNF was shown to confer behavioral improvements in a 6-hydroxydopamine rat model of PD [70,104]. On the other hand, it is reported that GDNF is not required for the survival of catecholaminergic neurons under physiological conditions [105]. The lack of GDNF in embryonic and adult mice using *loxP*/Cre (Nestin-Cre, AAV5-Cre, and Esr1-Cre) systems showed no significant differences in the number of TH-positive neurons, motor function, or metabolite levels of dopamine. In addition, it has been reported that overproduction of GDNF in the α-synuclein model does not prevent the aggregation of α-synuclein in the SN, loss of dopamine neurons by α-synuclein, or motor dysfunction [71,72]. Importantly, a potential cause of the non-response of GDNF in PD has been reported. In this study, GDNF signaling was blocked in dopaminergic neurons overexpressing α-synuclein in the rat SN. This block was accompanied by reduced expression of transcription factor Nurr1 and GDNF receptor Ret. Moreover, overexpression of Nurr1 restored the response to GDNF by α-synuclein [106]. However, more recent studies have not demonstrated RET downregulation in PD animal models or postmortem sporadic PD patient brains [107]. Another study confirmed that GDNF prevented the accumulation of misfolded α-synuclein in DA neuron cultures, and these effects were abolished by deletion of RET or inhibition of Src and Akt. Expression of constitutively active RET and GDNF-protected DA neurons from α-synuclein preformed fibrils-induced α-synuclein accumulation [108]. Trials investigating treatment of PD patients with GDNF have been in progress for 20 years. The initial research administered GDNF into the lateral ventricle of PD patients by the intracerebroventricular route [109]. In a multicenter, randomized, double-blind, placebo-controlled, sequential cohort study, 50 patients received GDNF monthly for 8 months and extended exposure up to an additional 20 months. However, no therapeutic effect was confirmed even at high concentrations of GDNF, and only side effects such as nausea, weight loss, and asymptomatic hyponatremia were reported. The lack of clinical effects in this trial was due to the possibility that GDNF was not effectively delivered into the target tissues, such as the putamen and SN, when administered intracerebroventricularly. This led to the consideration of other strategies for the administration of GDNF, and subsequent trials used intraparenchymal administration of GDNF. In an open-label study, five patients with PD were treated with intraputamenal infusion of GDNF [110]. Patients receiving GDNF showed a 30–60% improvement in the off-medication motor sub-score of the Unified Parkinson’s Disease Rating Scale (UPDRS) and a 61% improvement in the activities of daily living sub-score. Moreover, the beneficial effect of GDNF administration was confirmed by increased putamenal 18^F^-dopa uptake measured by positron emission tomography (PET). In addition, it was confirmed that TH-immunopositive nerve fibers were increased in the postmortem brain tissue study of a patient who had received unilateral infusion for 48 months [111]. In other open-label studies, unilateral intraputamenal infusions of GDNF were performed in 10 patients with PD, and motor dysfunction was improved without any serious side effects for 6 and 12 months [112,113]. Intraputamenal infusion of GDNF has shown promise for GDNF as a factor for clinical improvement in PD patients, and based on these results, a new clinical trial was conducted. GDNF was continuously administered into the posterior dorsal putamen using a chronic infusion pump, and the patient was evaluated for six months [114]. After six months, there was no difference from the placebo group, and some patients had problems developing neutralizing antibodies to GDNF. The study was withdrawn because of such stability issues; however, one patient experienced clinical improvement several years following GDNF treatment [115]. In another study, GDNF was administered intraputamenally every month for six months using a convection-enhanced delivery system [116,117]. There was no significant improvement in UPDRS score in GDNF-administered patients, but an increase in dopamine neuron function was confirmed by PET imaging. As a new strategy for delivering GDNF into target tissue, a method using viral vectors is currently being considered. AAV2, which enables long-term expression of transgenes without inducing an inflammatory response, is currently the vector of choice for clinical trials in PD patients. Currently, AAV2-GDNF is administered to the putamen, and treatment studies are ongoing (NCT04167540 and NCT01621581).

Various NTFs have been investigated in animal models for the treatment of other neurodegenerative diseases. ALS, the most common form of motor neuron disease, is a fatal adult-onset neurodegenerative disease that results in progressive and preferential degeneration and death of both the upper motor neurons of the motor cortex and the alpha lower motor neurons of the brain stem and spinal cord [118,119]. Several studies have documented the changes and neuroprotective effects of NTFs in ALS using in vivo and in vitro models. Expression levels of BDNF showed a significant reduction in spinal cord tissue obtained from SOD1 (G93A) mice, a murine model of ALS [67], and in lumbar spinal cord tissue of rat neonates that were injected intrathecally with CSF of ALS [68]. BDNF prevents cell death of motor neurons in the axotomized facial nucleus of the neonatal rat [120,121] and mediates antiapoptotic effects by the ERK and PI3K pathways [122]. Recently, studies have also confirmed that the modulation of TrkB via enhanced BDNF signaling increased neuronal survival in degenerating neurons in vitro [123] and improved motor dysfunction and motor neuron loss in ALS model mice [78]. GDNF has also been reported to have a protective effect on motor neurons in ALS. Disruption of the TNFR1-GDNF axis in astrocytes accelerates motor neuron degeneration and disease progression of ALS [75]. GDNF delivery prevents motor neurons from degenerating and preserves the axons that innervate the muscle; moreover, it has been shown to inhibit muscle atrophy in a transgenic mice model with the G93A human SOD1 mutation of ALS [76]. Furthermore, four-limb injection of AAV-GDNF in ALS mice postpones disease onset, delays progression of motor dysfunction, and prolongs life span [77]. However, despite the observation of these neuroprotective effects, there is also evidence that shows that therapeutic intervention of BDNF is unable to promote survival or prevent neuronal death in vivo. Many studies have shown that BDNF negatively affects motor neuron survival, which makes motor neurons more susceptible to damage [124,125]. Moreover, BDNF is effective in enhancing excitotoxic damage by enhancing glutamatergic activity in neurons [126]. Several studies have reported that BDNF plays a key role in motor neuron susceptibility to excitotoxicity [125,127,128]. In addition, muscles and CSF of patients with ALS exhibit elevated levels of BDNF [129] and GDNF [130]. These results suggest that the negative effects of NTFs also need to be considered for NTF therapeutic strategies for ALS.

Some NTFs such as BDNF, CNTF, and GDNF are promising candidates for future treatment as they affect the glial activation status and have a beneficial effect on the outcomes of neurodegenerative diseases. Exogenous BDNF treatment has been found to prevent the apoptotic cell death of astrocytes and neurons induced by neurotoxin 3-nitropropionic acid in culture conditions. BDNF treatment also regulates glutamate transporter expression and inflammatory responses in astrocytes. This observation suggests that the BDNF-induced beneficial function of astrocytes has therapeutic potential against neurodegenerative diseases [131]. Another study demonstrated that neuron-derived BDNF contributes to the regulation of synaptic density in the brain [132,133]. Exogenous CNTF increases the activation of astrocytes in the contralateral gray matter of the spinal cord when administered in a posttreatment manner after spinal cord injury (SCI) [134]. GDNF mRNA expression was found to be upregulated only in the astrocytes of the lesioned striatum in a 6-OHDA model of PD [135,136]. Another research showed that exogenous GDNF injections increased the number of phagocytic microglia in the spinal cord after SCI [137].

Taken together, evidence suggests that the absence of neurotrophic support contributes significantly to neurodegeneration and that NTFs have emerged as a promising therapeutic strategy for neurodegenerative diseases [138]. However, they have a relatively weak effect in the human clinical settings in contrast to their strong effect in animal models. Clinical trial design for investigating neuroprotection in neurodegenerative diseases remains challenging; however, inadequate designs may have resulted in failure to demonstrate neuroprotection. Therefore, preclinical work and careful consideration for all aspects of clinical trial design are required. There are several caveats that may affect test results and lead to test failure. Patients with neurodegenerative diseases have a wide variety of clinical symptoms and stages. Most clinical studies on NTFs observed changes in the CSF, plasma, or postmortem tissues during the late stage, so it is difficult to apply such changes to the early stages of the target tissue. Several studies of post-hoc analyses of trials have shown that clinical trials in patients at a late stage of disease have mostly failed, and patients with shorter illness duration or less severe symptoms have significant clinical benefit. These results suggest that patient selection and stratification have important implications for achieving clinical therapeutic effects with NTFs. Furthermore, more complex factors may be at play that inhibit the effects of NTFs in humans compared with that in animals. Animal models do not meticulously mimic the neurodegenerative processes and rate of disease progression in humans. For example, in addition to body weight, organ size and metabolic differences may also be the basis for the design of a clinical trial. Most importantly, protein delivery to the human brain has inherent difficulties, and it is probable that the low success rate of this approach is largely due to the protein not reaching the target at a sufficient concentration as well as off-target effects [139,140]. In addition, pharmacokinetic studies have shown that NTFs, such as BDNF [141] and CNTF [142], have a short half-life. This short half-life is a factor that further reduces the effectiveness of NTFs. CNTF and BDNF have extremely short half-lives of less than 2.9 min and 10 min, respectively, following intravenous injection into rodents. This short half-life is a factor that reduces the effectiveness of NTFs. The concentration of NTFs is also considered an important factor in clinical trials. Excessive upregulation of NTFs in the adult brain may contribute to epileptogenesis or induce an abnormal formation of synaptic networks [129,143,144]. These findings suggest that appropriate control of balance of NTFs is important for treating neurodegenerative diseases. Therefore, sustained expression of NTFs using an appropriate delivery system that protects the neurons in a specific target area is considered a viable therapeutic strategy for neurodegenerative disorders.

## 3. Rheb-mTORC1 Signaling against Neurodegenerative Diseases

Rheb is inactivated in the form of guanosine diphosphate-bound Rheb by a tuberous sclerosis complex (TSC) complex consisting of TSC1 (hamartin), TSC2 (tuberin), and TBC1D7 that act as guanosine triphosphatase (GTP)-activating proteins (GAT) [145]. Insulin and insulin-like growth factor 1 (IGF1), which activate G-protein-coupled and IGF1 receptors on target cell membranes, trigger the lipid kinase phosphatidylinositol-3 kinase (PI3K)-serine/threonine kinase Akt signaling pathway [146,147,148]. Activated Akt phosphorylates at the conserved consensus phosphorylation sequences of TSC2 and downregulates the GAP activity of the TCS complex [146]. Consequently, the activated form of GTP-bound Rheb increases and interacts with mTORC1 (Figure 1). It is well known that Rheb acts as a key activator of mTORC1, and its deletion can result in decreased cortical thickness and increased demyelination in the brain [37]. Accumulating evidence indicates that Rheb upregulation prevents neuronal loss and thus protects against neurodegenerative diseases, including AD [38,42], PD [73,149,150], and SCI [28].

Interestingly, in this context, our own recent studies have shown that a higher constitutive level of Rheb(S16H), a constitutively active mutant protein transduced using AAV serotype 1 (AAV1), can protect hippocampal neurons in the adult brain against neurotoxic conditions through neurotrophic interactions between neurons and astrocytes [39,40,41,151]. It is well known that mTORC1 acts as a downstream effector of Rheb. Rheb-mTORC1 signaling regulates various transcriptional and translational mechanisms associated with cell growth, proliferation, protein synthesis, and synaptic plasticity [152]. Previous studies have shown that the activation of Rheb upregulates the expression of NTFs such as BDNF, CNTF, and GDNF through the activation of mTORC1 in the adult brain. Moreover, several studies have demonstrated the role of the Rheb–mTOR axis in synaptic morphogenesis and axon elongation [153,154]. Akt–Rheb–mTORC1 signaling can enhance axon length and axon number per neuron as well as the number of neurons with multiple axons [155]. Furthermore, learning, memory, and long-term synaptic plasticity depend on de novo protein synthesis. The activation of the Rheb–mTOR axis acts as an important signaling pathway for the maintenance of long-term potentiation (LTP) and learning and memory functions [32,156]. It has been reported that Rheb interacts with beta-secretase 1 (BACE1) to inhibit BACE1 activity in the hippocampus of mouse brain, leading to the reduction of Aβ generation [38]. In the brains of patients with AD, it was shown that the expression of Rheb was significantly downregulated compared to that a normal healthy brain [157]. Depletion of Rheb can lead to the behavioral hallmarks of AD progression, such as defects in spatial memory functions [158].

In PD conditions, the role of mTOR remains controversial as it could exert either neuroprotective [159,160] or neurotoxic [161,162] effects in different PD models. Evidence accumulated in recent years has shown that mTOR signaling is altered during PD progression [163]. Various studies have demonstrated that 1-methyl-4-phenyl-1,2,3,6-tetrahydropyridine (MPTP) and 6-hydroxydopamine (6-OHDA) suppress mTOR signaling and induce neuronal cell death [164,165,166,167,168,169]. Moreover, inhibition of mTORC1 by rapamycin can induce neuronal cell death in oxidative stress-induced condition [170], and regulated in DNA damage and development 1 (REDD1), an intrinsic repressor of mTORC1, was found to be highly expressed in a neurotoxin-induced cell model and in the substantia nigra of patients with PD [171]. Interestingly, one study showed that the activation of Akt/mTOR signaling in the dopaminergic neurons of the substantia nigra mediated axon protection in the neurotoxin lesions of a 6-OHDA-induced model [172], suggesting the importance of the Akt/Rheb/mTOR axis to protect axons during PD development. On the other hand, several studies have reported negative roles of mTOR, such as increasing the concentration of α-synuclein in PD. Intracellular accumulation of α-synuclein in structures known as Lewy bodies is a hallmark of PD and has been implicated in the pathogenesis of sporadic and familial PD [173,174,175]. It is known that clearance of α-synuclein occurs through the process of autophagy, and mTOR acts as a negative regulator of this process [176]. In the α-synuclein animal model, mTOR activity is increased, whereas the autophagy pathway is inactive [161,177]. These changes are associated with neurodegeneration in PD. The autophagy process occurs through the activation of an initiator called Unc51-like kinase 1 (ULK1) and the ATG complex [178]. mTOR inhibits the initiation of autophagy through the phosphorylation of the P757 site of ULK1. The reduction of autophagy by mTOR has been shown to be restored by rapamycin, an mTOR inhibitor, in an α-synuclein animal model [161,177]. Given the importance of mTOR in the clearance of α-synuclein, inhibition of mTOR signaling may be a viable therapeutic strategy. However, mTOR signaling regulates several essential cellular functions, including synaptic plasticity, memory formation, maintenance of nerve cells, and regulating NTF expression. Therefore, because mTOR is essential for cell survival and growth, maintaining an appropriate level of mTOR activity is vital.

Overall, the evidence reported to date suggests that Rheb–mTOR signaling is associated with neuronal loss that occurs in different neurodegenerative diseases and could therefore be a potential therapeutic target for a wide range of such disorders.

## 4. Therapeutic Potential of Rheb(S16H) Transduction via AAV1 against Neurodegenerative Diseases

As described earlier, the Rheb–mTOR signaling pathway is a key regulator of diverse mechanisms involved in the survival and regeneration of neurons. Accordingly, several studies have identified Rheb mutations that cause this protein to become constitutively active [73,151]. Our previous studies also demonstrated that the substitution of serine by histidine at position 16 of Rheb [Rheb(S16H)] significantly induced the expression of NTFs, such as GDNF, CNTF, and BDNF in the adult brain [39,42,69,73,74,179,180]. Our studies further showed that the upregulation of neuronal Rheb(S16H) protects neurons from neuronal degeneration and promotes axonal regrowth in the hippocampus and the nigrostriatal dopamine system of the adult brain [41,73,74,149,150,180]. Our laboratory research further demonstrated that the activation of mTORC1 induced by Rheb(S16H) transduction in hippocampal neurons led to BDNF production, which protected the rat hippocampus against thrombin-induced neurotoxicity [40]. Similarly, Rheb(S16H) transduction using AAV1 viral vector in the hippocampus of transgenic 5XFAD mice, a transgenic mouse model of AD carrying five mutations associated with early-onset familial AD, was found to prevent against cognitive function impairment [39,42]. To date, the neuronal upregulation of Rheb(S16H) has demonstrated several neuroprotective effects in the hippocampus of the rat brain under neurotoxic conditions. Our previous studies have demonstrated that AAV1-Rheb(S16H) transduction in the hippocampal neurons prevents thrombin-induced neuronal cell death in the hippocampus of the rat brain [39,40]. More recently, AAV1-Rheb(S16H) transduction in our laboratory was found to have preventive effects against cognitive decline and LTP impairment in 5XFAD mice [42]. These beneficial effects could be mediated by the stimulation of neuronal BDNF production through the activation of the Rheb-mTORC1 signaling pathway after AAV1-Rheb(S16H) transduction, which can occur irrespective of the levels of neuroinflammatory molecules. Moreover, we recently found that Rheb(S16H) transduction using AAV1 viral vector in the hippocampal neurons increased the expression of GDNF in both neurons and astrocytes via autocrine and paracrine BDNF-TrkB signaling [39]. 

Similarly, in our other previous studies, we found that the expression of neuronal Rheb(S16H) protects dopaminergic neurons through the upregulation of NTF expression in the substantia nigra in a neurotoxin-induced PD model. AAV1-hRheb(S16H) transduction of dopamine neurons protects against 6-OHDA-induced neurodegeneration and induces neuroprotective effect, including the abilities to both preserve and restore the dopaminergic axonal projections, through mTORC1 signaling [149,150] Other studies have demonstrated that the transduction of dopaminergic neurons by AAV1-hRheb(S16H) induced the upregulation of both CNTF and CNTFRα, which contributed to protection against 1-methyl-4-phenylpyridinium (MPP^+^)-induced neurotoxicity in nigrostriatal striatal dopaminergic projections [73]. In addition, Rheb(S16H) expression induced an increase in the levels of the phospho-cyclic adenosine monophosphate response element-binding protein (p-CREB) in dopaminergic neurons, which may be involved in the protective effects through the production of GDNF and BDNF [74]. Neurotoxic inflammation mediated by glial cells is considered as an important mechanism in the pathogenesis of PD and can exacerbate the cell loss of dopaminergic neurons and worsen the symptoms of the disease [96,181]. Our recent observations showed that Rheb(S16H) transduction in dopamine neurons played a role in the neuroprotection of the nigrostriatal dopamine system by inducing mTORC1 activation and BDNF/GDNF production without controlling neuroinflammation under pKr-2 (endogenous microglia activator)-induced neurotoxic inflammatory environment [41] (Figure 2).

In the hippocampus of the adult brain, AAV1-Rheb(S16H) transduction was found to induce a persistent increment in the expression levels of TrkB and CNTFRα within activated astrocytes and hippocampal neurons, respectively [39]. We also found that neuronal BDNF produced by AAV1-Rheb(S16H) transduction in hippocampal neurons induced the activation of astrocytes and mediated the production of CNTF through the activation of astrocytic TrkB. Therefore, the upregulation of neuronal BDNF and astrocytic CNTF by hRheb(S16H) transduction could have a positive synergistic effect on the survival of hippocampal neurons in vivo [39]. In our more recent analyses, AAV1-Rheb(S16H) transduction in 5XFAD mice was found to exert preventive effects against LTP impairment and cognitive decline [42]. These beneficial effects may be attributable to the interaction of multiple NTFs between neurons and astrocytes produced by the Rheb(S16H) transduction in hippocampal neurons, thereby resulting in neuroprotection in the hippocampus of 5XFAD mice (Figure 2).

## 5. Conclusions

Although a cure for neurodegenerative diseases has yet to be developed, there are several symptomatic treatments. Remarkably however, all these diseases will eventually progress. Our previous studies have clarified that the transduction of hippocampal neurons by Rheb(S16H) transduction stimulates NTF production, thereby strengthening the neuroprotective system and reducing neurodegeneration in the hippocampus and dopaminergic systems of the adult brain. Therefore, although further research is necessary to determine the clinical applicability of AAV1-Rheb(S16H) transduction to treat neurodegenerative diseases, this approach may be a useful strategy to protect neurons in the lesioned brain, and it may have beneficial effects in patients with neurological disorders, such as AD and PD.

## Figures and Tables

**Figure 1 ijms-22-03064-f001:**
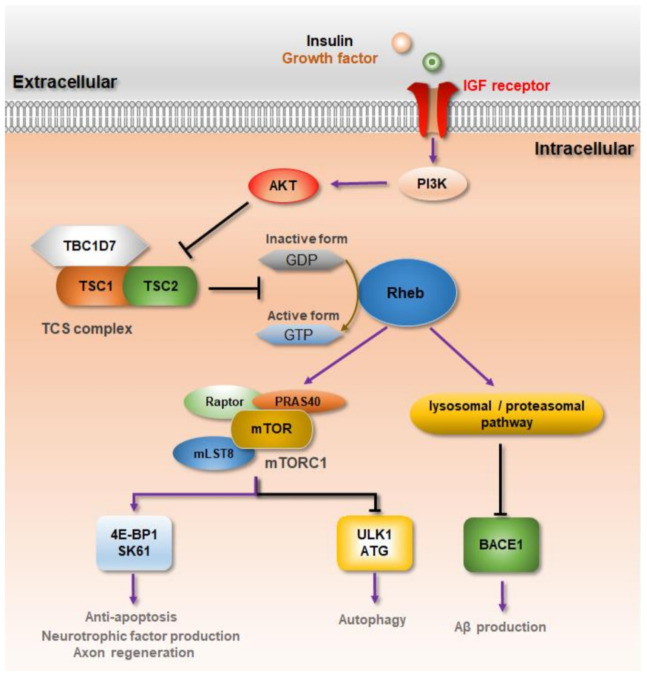
Rheb-mTORC1 signaling pathway in neurodegeneration. Binding of growth factors, such as insulin or insulin-like growth factor (IGF), to receptors stimulates PI3K signaling. The activity of PI3K-AKT mediates the activation of Rheb-mTORC1 by disinhibiting the Rheb-inhibiting TCS complex. Fully activated Rheb-mTORC1 phosphorylates S6K-1 or 4E-BP1 activate the protein of the translation machinery. In neurodegenerative diseases, Rheb-mTORC1 has negative effects, such as inhibition of autophagy, but promotes cell growth, regeneration, and neuroprotection by regulating the expression of neurotrophic factors, exon regeneration, and anti-apoptosis. In addition, Rheb inhibits Aβ formation through the induction of lysosomal and proteasomal degradation by binding to beta-secretase 1 (BACE) in Alzheimer’s disease (AD) conditions.

**Figure 2 ijms-22-03064-f002:**
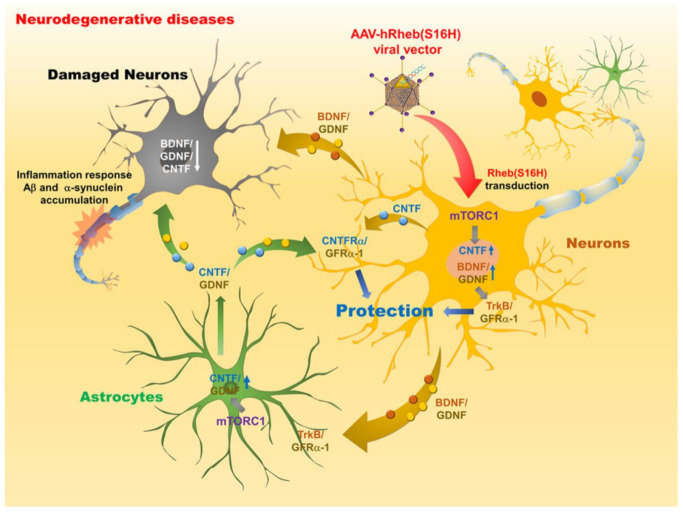
Schematic representation of the mechanisms of a neuroprotective system following AAV-hRheb(S16H) transduction in neurodegenerative diseases. In neurodegenerative diseases, various neurotoxicities, including Aβ deposition, α-synuclein aggregation, and inflammatory responses, induce axon loss and demyelination of specific neurons as well as reduction of neurotrophic factors (NTFs), leading to decreased memory/cognition and movement decline. hRheb(S16H) transduction of hippocampal or SNpc neurons using adeno-associated virus serotype 1 (AAV1) can induce the activation of mTORC1, which in turn stimulates the production of NTFs, such as brain-derived neurotrophic factor (BDNF), glial cell line-derived neurotrophic factor (GDNF), and ciliary neurotrophic factor (CNTF) in the neurons. Increased BDNF and GDNF expressions contribute to neuroprotection through the activation of TrkB/GFRα-1 receptor, in the hippocampus and SNpc of brain, respectively. Moreover, CNTF production by hRheb(S16H) expression in SNpc neurons mediates neuroprotective effects through the CNTFRα receptor. In addition, BDNF/GDNF expression induced in hRheb(S16H)-expressing neurons result in functional interactions between neurons and astrocytes in the hippocampus, leading to the production of astrocytic CNTF and GDNF for hippocampal protection.

**Table 1 ijms-22-03064-t001:** Vulnerable neuronal populations in various neurodegenerative diseases.

Neurodegenerative Diseases	Target System	Target Tissues	References
Alzheimer’s Disease	Cholinergic Neurons	Nucleus basalis (NB)	[43]
	Noradrenergic Neurons	Locus coeruleus (LC)	[44]
	Dopaminergic Neurons	Substantia nigra (SN)	[45]
	Serotonergic Neurons	Dorsal raphe nucleus (DRN)	[46]
	Adrenergic Neurons	Rostral ventral lateral medulla C-1 neurons	[47]
Parkinson’s Disease	Cholinergic Neurons	Nucleus basalis (NB)	[50]
	Noradrenergic Neurons	Locus coeruleus (LC)	[52]
	Dopaminergic Neurons	Substantia nigra (SN)	[49]
	Serotonergic Neurons	Dorsal raphe nucleus (DRN)	[48]
	Adrenergic Neurons	Rostral ventral lateral medulla C-1 neurons	[51]
Huntington’s Chorea	Dopaminergic Neurons	Striatum	[56]
	Cholinergic Neurons	Thalamostriatal axodendritic terminals	[53]
	GABAergic Neurons	striatum	[54,55]
	Glutamate Neurons	Striatum	[57]

**Table 2 ijms-22-03064-t002:** Animal model used to study the therapeutic effect of neurotrophic factors in various neurodegenerative diseases.

Diseases	Animal Model	Neurotrophic Factor	Effect	Reference
Alzheimer’s Disease	J20 (human APP mutant)	BDNF	Improve	[21]
	Tg2576	CNTF	Improve	[22]
	Thrombin	BDNF CNTF	Improve	[39,40]
	5XFAD	BDNF CNTF	Improve	[42]
	P301L	BDNF	Improve	[60]
	Thrombin	GDNF	Improve	[69]
Parkinson’s Disease	Inflammation (pKr-2)	GDNF BDNF	Improve	[41]
	Gdnf(+/−)	GDNF	Improve	[63]
	6-OHDA	GDNF	Improve	[70]
	α-synuclein	GDNF	Worsen	[71,72]
	MPP^+^	CNTF	Improve	[73]
	MPP^+^	GDNF BDNF	Improve	[74]
Huntington’s Disease	bdnf(+/−)	BDNF	Improve	[65]
Amyotrophic lateral sclerosis	SOD1(G93A)	GDNF	Improve	[75,76,77]
		BDNF	Improve	[78]

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
