# Peer review of "Therapeutic Potential of AAV1-Rheb(S16H) Transduction against Neurodegenerative Diseases"

_ijms, 2021, doi:10.3390/ijms22063064_

Round 1
Reviewer 1 Report
Reviewer 1:
The manuscript „Therapeutic potential of AAV1-Rheb(S16H) transduction against neurodegenerative Diseases” discusses very interesting and potentially effective approach to neuroprotection in neurodegenerative diseases by expression of Rheb(S16H). The authors convincingly describe evidence, and also frame mechanism of Rheb action in the intensively explored field of neurotrophic factors in neurodegenerative diseases, however this is not done in sufficient depth. Since neurotrophic factors are critical for AAV1-Rheb(S16H) mechanism of action, their role in disease, therapeutic potential – should be more thoroughly discussed. This part should be either expanded, or at least mention also some potential caveats and ongoing discussion in the field to show to the reader that neurotrophic factors role in neurodegenerative disease progression and treatment development require some caution.
Specifically :
- Page 3: „loss of selectively vulnerable populations…” could the authors list those? Again on page 5 there are “specific cell populations” , they should be listed.
- Page 3: “Numerous studies have also demonstrated that a decrease in the levels of neurotrophic factors (NTFs), such as brain-derived neurotrophic factor (BDNF) and ciliary neurotrophic factor (CNTF), is associated with the pathology of neurodegenerative diseases, and that this decrease is closely linked to neuronal cell death [15-18].” The cited papers are reviews or don’t directly address problem of decreased in levels of NTFs in neurodegeneration, please support your thesis with correct original research papers (such papers are later cited in similar context on page 5 (citations 37-42). Furthermore please some caveats should be discusses there. Decrease in neurotrophic factors in neurodegenerative disease in humans (in contrast to animal models) usually use proxy measurements of NTFs levels in CSF or plasma, or late-stage post mortem tissues drawing conclusions from which might not be applicable to early stage patients. At the same time data from animal studies rely on very imperfect models, as etiological validity of neurodegenerative disorders models is quite poor (especially for AD and PD). Furthermore results even from such imperfect studies have not been equivocal. I actually agree with the authors that there is probably decrease in NTFs, but in a review paper it should be discussed that this is not black and white picture.
- Main GDNF receptor is RET tyrosine kinase together with GFRa1 not GFRa1 alone.
- Same as in point 3 – on page 6 GFRa1 is mentioned as GDNF receptor, please mention that it’s coreceptor together with RET.
- It’s great that authors consider not only dopaminergic but also other types of neurons in PD. However they could mention that dopaminergic degeneration seem to be leading cause of motor dysfunctions and main target of existing and actively developed therapeutics (albeit degeneration in other system probably don’t get deserved attentions, so it’s really great that authors consider it)
- On page 7 It should be mentioned that GDNF was tested in clinical trials, however with mixed results, see:
- Paul G, Sullivan AM. Trophic factors for Parkinson’s disease: Where are we and where do we go from here? Eur J Neurosci. 2018;(June 2018):440–52.
- Nasrolahi A, Mahmoudi J, Akbarzadeh A, Karimipour M, Sadigh-Eteghad S, Salehi R, et al. Neurotrophic factors hold promise for the future of Parkinson’s disease treatment: Is there a light at the end of the tunnel? Rev Neurosci. 2018;29(5):475–89.
- Chmielarz P, Saarma M. Neurotrophic factors for disease-modifying treatments of Parkinson’s disease: gaps between basic science and clinical studies. Pharmacol Reports. 2020;
- Also, regarding GDNF in PD, there is ongoing dispute regarding requirement of GDNF for maintenance of adult dopaminergic neurons, please consider this, see:
- Pascual A, Hidalgo-Figueroa M, Piruat JI, Pintado CO, Gómez- Díaz R, López-Barneo J. Absolute requirement of GDNF for adult catecholaminergic neuron survival. Nat Neurosci. 2008;11:755–61. https ://doi.org/10.1038/nn.2136.
- Kopra J, Vilenius C, Grealish S, Härma MA, Varendi K, Lind- holm J, Castrén E, Võikar V, Björklund A, Piepponen TP, Saarma MAJ. GDNF is not required for catecholaminergic neu- ron survival in vivo. Nat Neurosci. 2015;18:319–22. https ://doi. org/10.1038/nn.3941.
- Enterría-Morales D, López-López I, López-Barneo J, d’Anglemont de Tassigny X (2020) Role of glial cell line-derived neurotrophic factor in the maintenance of adult mesencephalic catecholaminergic neurons. Mov Disord. https ://doi.org/10.1002/ mds.27986
- Moreover there were reports that GDNF is ineffective to protect against alpha-synuclein (See
- Decressac M, Ulusoy A, Mattsson B, Georgievska B, Romero- Ramos M, Kirik D, et al. GDNF fails to exert neuroprotec- tion in a rat α-synuclein model of Parkinson’s disease. Brain. 2011;134:2302–11. https ://doi.org/10.1093/brain /awr14 9.
- Bianco C, Déglon N, Pralong W, Aebischer P. Lentiviral nigral delivery of GDNF does not prevent neurodegeneration in a genetic rat model of Parkinson’s disease. Neurobiol Dis. 2004;17:283–9. https ://doi.org/10.1016/j.nbd.2004.06.008.
- Decressac M, Kadkhodaei B, Mattsson B, Laguna A, Perlmann T, Björklund A, et al. Synuclein-induced down-regulation of Nurr1 disrupts GDNF signaling in nigral dopamine neurons. Sci Transl Med. 2012;4:163ra156–ra156. https ://doi.org/10.1126/ scitr anslm ed.30046 76.
However this was later challenged, see:
- Su X, Fischer DL, Li X, Bankiewicz K, Sortwell CE, Federoff HJ. Alpha-synuclein mRNA is not increased in sporadic PD and alpha-synuclein accumulation does not block GDNF signaling in Parkinson’s disease and disease models. Mol Ther. 2017. https :// doi.org/10.1016/j.ymthe .2017.04.018.
- Chmielarz P. GDNF / RET Signaling Pathway Activation Eliminates Lewy Body Pathology in Midbrain Dopamine Neurons. Mov Disord. 2020;1:1–12.
- at page 8 authors write about problems with protein delivery and failed clinical trials of NTFs, but only briefly. This topic should somewhat expanded. Failed clinical trials are strong argument against neurotrophic factors in neurodegenerative disease treatment, and in fact greatly hindered development of the field. There are multiple factors which probably contributed to their failure (see mentioned in point 6 reviews). The authors should more convincingly rebuke arguments that NTFs / neurotrophic factors are not working in neurodegeneration in humans by expanding this sections.
- The manuscript would greatly benefit of additional figure with intracellular signaling pathways linked to Rheb (as described on page 8-9). The figure could also contain schema of downstream intracellular processes regulated by Rheb-mTOR pathway (relevant for neurodegeneration)
- There mounting evidence of importance of alpha-synuclein in PD. Could the authors discus protective potential of Akt/Rheb/mTOR pathway in this context?
Author Response
Review 1
The manuscript „Therapeutic potential of AAV1-Rheb(S16H) transduction against neurodegenerative Diseases” discusses very interesting and potentially effective approach to neuroprotection in neurodegenerative diseases by expression of Rheb(S16H). The authors convincingly describe evidence, and also frame mechanism of Rheb action in the intensively explored field of neurotrophic factors in neurodegenerative diseases, however this is not done in sufficient depth. Since neurotrophic factors are critical for AAV1-Rheb(S16H) mechanism of action, their role in disease, therapeutic potential–should be more thoroughly discussed. This part should be either expanded, or at least mention also some potential caveats and ongoing discussion in the field to show to the reader that neurotrophic factors role in neurodegenerative disease progression and treatment development require some caution.
Specifically:
- Comment: Page 3: „loss of selectively vulnerable populations…” could the authors list those? Again on page 5 there are “specific cell populations” , they should be listed.
Response: We appreciate the reviewer’s comment. A list of vulnerable populations in neurodegenerative diseases have been added to Table 1.
- Comment: Page 3: “Numerous studies have also demonstrated that a decrease in the levels of neurotrophic factors (NTFs), such as brain-derived neurotrophic factor (BDNF) and ciliary neurotrophic factor (CNTF), is associated with the pathology of neurodegenerative diseases, and that this decrease is closely linked to neuronal cell death [15-18].” The cited papers are reviews or don’t directly address problem of decreased in levels of NTFs in neurodegeneration, please support your thesis with correct original research papers (such papers are later cited in similar context on page 5 (citations 37-42). Furthermore please some caveats should be discusses there. Decrease in neurotrophic factors in neurodegenerative disease in humans (in contrast to animal models) usually use proxy measurements of NTFs levels in CSF or plasma, or late-stage post mortem tissues drawing conclusions from which might not be applicable to early stage patients. At the same time data from animal studies rely on very imperfect models, as etiological validity of neurodegenerative disorders models is quite poor (especially for AD and PD). Furthermore results even from such imperfect studies have not been equivocal. I actually agree with the authors that there is probably decrease in NTFs, but in a review paper it should be discussed that this is not black and white picture.
Response: We thank the reviewer for their comment. As recommended by the reviewer, we have replaced the papers mentioned in the comments with original research papers (pages 3 and 6 in the revised manuscript).
The limitations of NTF change studies in patients with neurodegenerative diseases, and the problems with animal models not fully reflecting human disease have been added to the discussions regarding failures of clinical trials. (pages 12–13 in the revised manuscript).
We have added text describing that appropriate control of NTF balance in neurodegenerative diseases is important. We have also added that abnormal effects occur when the expression of NTF increases in diseases, such as seizures, and that negative regulation occurs with increased mTOR expression in PD.
Page 13 in the revised manuscript: The concentration of NTFs is also considered an important factor in clinical trials. Excessive upregulation of NTFs in the adult brain may contribute to epileptogenesis or induce an abnormal formation of synaptic networks [126,140,141]. These findings suggest that appropriate control of balance of NTFs is important for treating neurodegenerative diseases.
Pages 15-16 in the revised manuscript: several studies have reported negative roles of mTOR, such as increasing the concentration of α-synuclein in PD. Intracellular accumulation of α-synuclein in structures known as Lewy bodies is a hallmark of PD and has been implicated in the pathogenesis of sporadic and familial PD [171-173]. It is known that clearance of α-synuclein occurs through the process of autophagy, and mTOR acts as a negative regulator of this process [174]. In the α-synuclein animal model, mTOR activity is increased, whereas the autophagy pathway is inactive [159,175]. These changes are associated with neurodegeneration in PD. The autophagy process occurs through the activation of an initiator called Unc51-like kinase 1 (ULK1) and the ATG complex [176]. mTOR inhibits the initiation of autophagy through the phosphorylation of the P757 site of ULK1. The reduction of autophagy by mTOR has been shown to be restored by rapamycin, an mTOR inhibitor, in an α-synuclein animal model [159,175]. Given the importance of mTOR in the clearance of α-synuclein, inhibition of mTOR signaling may be a viable therapeutic strategy. However, mTOR signaling regulates several essential cellular functions, including synaptic plasticity, memory formation, maintenance of nerve cells, and regulating NTF expression. Therefore, because mTOR is essential for cell survival and growth, maintaining an appropriate level of mTOR activity is vital.
- Comment: Main GDNF receptor is RET tyrosine kinase together with GFRa1 not GFRa1 alone.
Response: As in our response to comment 4, we have added text about RET tyrosine kinase to the revised manuscript as shown below.
Pages 6-7 in the revised manuscript: In addition, it was reported that rearranged during transfection (RET) tyrosine kinase, which acts as a co-receptor of GDNF alongside GFRa1, was decreased in hippocampal neurons in a chronic cerebral hypoperfusion model of dementia [75]; moreover, overexpression of RET in AD neurons was related to neuronal survival [56,76]
- Comment: Same as in point 3on page 6 GFRa1 is mentioned as GDNF receptor, please mention that it’s coreceptor together with RET.
Response: As suggested by the reviewer, we have added a description of RET. We explain that it acts as a co-receptor of GDNF alongside with GFR1 and is altered in the brain of patients with neurodegenerative diseases (page 6 in the revised manuscript).
- Comment: It’s great that authors consider not only dopaminergic but also other types of neurons in PD. However they could mention that dopaminergic degeneration seem to be leading cause of motor dysfunctions and main target of existing and actively developed therapeutics (albeit degeneration in other system probably don’t get deserved attentions, so it’s really great that authors consider it)
Response: We thank the reviewer for their positive comment. We have revised the text as below.
Page 7 in the revised manuscript: PD is pathologically characterized by the progressive death of heterogeneous populations of neurons, including dopaminergic, cholinergic, noradrenergic, and serotonergic neurons [79-85]. However, dopaminergic degeneration is considered a major cause of motor dysfunction and is the main target of existing and actively developed therapeutics.
- Comment: On page 7 It should be mentioned that GDNF was tested in clinical trials, however with mixed results, see:
Paul G, Sullivan AM. Trophic factors for Parkinson’s disease: Where are we and where do we go from here? Eur J Neurosci. 2018;(June 2018):440–52.
Nasrolahi A, Mahmoudi J, Akbarzadeh A, Karimipour M, Sadigh-Eteghad S, Salehi R, et al. Neurotrophic factors hold promise for the future of Parkinson’s disease treatment: Is there a light at the end of the tunnel? Rev Neurosci. 2018;29(5):475–89.
Chmielarz P, Saarma M. Neurotrophic factors for disease-modifying treatments of Parkinson’s disease: gaps between basic science and clinical studies. Pharmacol Reports. 2020;
Response: We thank the reviewer for their valuable comment. We have checked the clinical trials that used GDNF in patients with Parkinson’s disease and have added the content of several relevant clinical studies to the revised manuscript (pages 9–10 in the revised manuscript).
- Comment: Also, regarding GDNF in PD, there is ongoing dispute regarding requirement of GDNF for maintenance of adult dopaminergic neurons, please consider this, see:
Pascual A, Hidalgo-Figueroa M, Piruat JI, Pintado CO, Gómez- Díaz R, López-Barneo J. Absolute requirement of GDNF for adult catecholaminergic neuron survival. Nat Neurosci. 2008;11:755–61. https ://doi.org/10.1038/nn.2136.
Kopra J, Vilenius C, Grealish S, Härma MA, Varendi K, Lind- holm J, Castrén E, Võikar V, Björklund A, Piepponen TP, Saarma MAJ. GDNF is not required for catecholaminergic neu- ron survival in vivo. Nat Neurosci. 2015;18:319–22. https ://doi. org/10.1038/nn.3941.
Enterría-Morales D, López-López I, López-Barneo J, d’Anglemont de Tassigny X (2020) Role of glial cell line-derived neurotrophic factor in the maintenance of adult mesencephalic catecholaminergic neurons. Mov Disord. https ://doi.org/10.1002/ mds.27986
Response: We have added further literature on the role that GDNF plays in regard to dopamine neurons in PD and related physiological conditions. We have also added a discussion on the results of the current controversial study of GDNF to the revised manuscript (pages 8–10 in the revised manuscript).
- Comment: Moreover there were reports that GDNF is ineffective to protect against alpha-synuclein (See
Decressac M, Ulusoy A, Mattsson B, Georgievska B, Romero- Ramos M, Kirik D, et al. GDNF fails to exert neuroprotec- tion in a rat α-synuclein model of Parkinson’s disease. Brain. 2011;134:2302–11. https ://doi.org/10.1093/brain /awr14 9.
Bianco C, Déglon N, Pralong W, Aebischer P. Lentiviral nigral delivery of GDNF does not prevent neurodegeneration in a genetic rat model of Parkinson’s disease. Neurobiol Dis. 2004;17:283–9. https ://doi.org/10.1016/j.nbd.2004.06.008.
Decressac M, Kadkhodaei B, Mattsson B, Laguna A, Perlmann T, Björklund A, et al. Synuclein-induced down-regulation of Nurr1 disrupts GDNF signaling in nigral dopamine neurons. Sci Transl Med. 2012;4:163ra156–ra156. https ://doi.org/10.1126/ scitr anslm ed.30046 76.
However this was later challenged, see:
Su X, Fischer DL, Li X, Bankiewicz K, Sortwell CE, Federoff HJ. Alpha-synuclein mRNA is not increased in sporadic PD and alpha-synuclein accumulation does not block GDNF signaling in Parkinson’s disease and disease models. Mol Ther. 2017. https :// doi.org/10.1016/j.ymthe .2017.04.018.
Chmielarz P. GDNF / RET Signaling Pathway Activation Eliminates Lewy Body Pathology in Midbrain Dopamine Neurons. Mov Disord. 2020;1:1–12.
Response: We appreciate the important comments made by the reviewer. We have added a discussion on the disputed findings of GDNF in PD related to alpha-synuclein as well as further literature that identified the potential cause of ineffectiveness of GDNF in PD to the revised manuscript (pages 8–10 in the revised manuscript).
- Comment: at page 8 authors write about problems with protein delivery and failed clinical trials of NTFs, but only briefly. This topic should somewhat expanded. Failed clinical trials are strong argument against neurotrophic factors in neurodegenerative disease treatment, and in fact greatly hindered development of the field. There are multiple factors which probably contributed to their failure (see mentioned in point 6 reviews). The authors should more convincingly rebuke arguments that NTFs / neurotrophic factors are not working in neurodegeneration in humans by expanding this sections.
Response: We appreciate the reviewers’ valuable comments. We have further considered the problem of therapeutic efficacy of NTF in animal studies showing poor results in human clinical trials. We have added text accordingly to the revised manuscript. We have described it as shown below.
Pages 12-13 in the revised manuscript: Taken together, evidence suggests that the absence of neurotrophic support contributes significantly to neurodegeneration and that NTFs have emerged as a promising therapeutic strategy for neurodegenerative diseases [135]. However, they have a relatively weak effect in the human clinical settings in contrast to their strong effect in animal models. Clinical trial design for investigating neuroprotection in neurodegenerative diseases remains challenging; however, inadequate designs may have resulted in failure to demonstrate neuroprotection. Therefore, preclinical work and careful consideration for all aspects of clinical trial design are required. There are several caveats that may affect test results and lead to test failure. Patients with neurodegenerative diseases have a wide variety of clinical symptoms and stages. Most clinical studies on NTFs observed changes in the CSF, plasma, or postmortem tissues during the late stage, so it is difficult to apply such changes to the early stages of the target tissue. Several studies of post-hoc analyses of trials have shown that clinical trials in patients at a late stage of disease have mostly failed, and patients with shorter illness duration or less severe symptoms have significant clinical benefit. These results suggest that patient selection and stratification have important implications for achieving clinical therapeutic effects with NTFs. Furthermore, more complex factors may be at play that inhibit the effects of NTFs in humans compared with that in animals. Animal models do not meticulously mimic the neurodegenerative processes and rate of disease progression in humans. For example, in addition to body weight, organ size and metabolic differences may also be the basis for the design of a clinical trial. Most importantly, protein delivery to the human brain has inherent difficulties, and it is probable that the low success rate of this approach is largely due to the protein not reaching the target at a sufficient concentration as well as off-target effects [136,137]. In addition, pharmacokinetic studies have shown that NTFs, such as BDNF [138] and CNTF [139], have a short half-life. This short half-life is a factor that further reduces the effectiveness of NTFs. CNTF and BDNF have extremely short half-lives of less than 2.9 minutes and 10 minutes, respectively, following intravenous injection into rodents. This short half-life is a factor that reduces the effectiveness of NTFs. The concentration of NTFs is also considered an important factor in clinical trials. Excessive upregulation of NTFs in the adult brain may contribute to epileptogenesis or induce an abnormal formation of synaptic networks [126,140,141]. These findings suggest that appropriate control of balance of NTFs is important for treating neurodegenerative diseases. Therefore, sustained expression of NTFs using an appropriate delivery system that protects the neurons in a specific target area is considered a viable therapeutic strategy for neurodegenerative disorders.
- Comment: The manuscript would greatly benefit of additional figure with intracellular signaling pathways linked to Rheb (as described on page 8-9). The figure could also contain schema of downstream intracellular processes regulated by Rheb-mTOR pathway (relevant for neurodegeneration)
Response: As suggested by the reviewer, we have added a schematic that includes the intracellular signaling pathway of Rheb and the downstream intracellular processes regulated by the Rheb-mTOR pathway. This additional figure is shown in Figure 1 of the revised manuscript.
There mounting evidence of importance of alpha-synuclein in PD. Could the authors discus protective potential of Akt/Rheb/mTOR pathway in this context?
Response: We thank the reviewer for their comments and advice. In accordance with the comments, we have added a description of the role of the Akt/Rheb/mTOR pathway in alpha-synuclein pathology to the relevant section, as below.
Pages 15-16 in the revised manuscript: several studies have reported negative roles of mTOR, such as increasing the concentration of α-synuclein in PD. Intracellular accumulation of α-synuclein in structures known as Lewy bodies is a hallmark of PD and has been implicated in the pathogenesis of sporadic and familial PD [171-173]. It is known that clearance of α-synuclein occurs through the process of autophagy, and mTOR acts as a negative regulator of this process [174]. In the α-synuclein animal model, mTOR activity is increased, whereas the autophagy pathway is inactive [159,175]. These changes are associated with neurodegeneration in PD. The autophagy process occurs through the activation of an initiator called Unc51-like kinase 1 (ULK1) and the ATG complex [176]. mTOR inhibits the initiation of autophagy through the phosphorylation of the P757 site of ULK1. The reduction of autophagy by mTOR has been shown to be restored by rapamycin, an mTOR inhibitor, in an α-synuclein animal model [159,175]. Given the importance of mTOR in the clearance of α-synuclein, inhibition of mTOR signaling may be a viable therapeutic strategy. However, mTOR signaling regulates several essential cellular functions, including synaptic plasticity, memory formation, maintenance of nerve cells, and regulating NTF expression. Therefore, because mTOR is essential for cell survival and growth, maintaining an appropriate level of mTOR activity is vital.
Reviewer 2 Report
In this article, Nam and collaborators review the involvement of neurotrophic factors in certain neurodegenerative diseases and the importance of targeting them in order to treat those diseases. Moreover, they talk about the use of a modified version of the Rheb protein delivered by AAV1 vectors to target the mTOR pathway and to enhance the expression of neurotrophic factors.
The review is comprehensive and very well-organized. However, some major and minor points should be addressed before being accepted.
Major point:
- Some parts of the manuscript, such as “The importance of neurotrophic factor therapeutic strategy for neurodegenerative diseases” or “Rheb-mTORC1 signalling pathway as a potential therapeutic target for neurodegenerative diseases” should be re-written with a more appropriate and clear English style.
Minor points:
- Please, carefully proof-read the entire manuscript to eliminate some grammatical errors and to avoid some word repetitions. It should be desirable to use more synonyms.
- It is not clear in the introduction whether Rheb can be activated by NTFs (line 6 of the second paragraph) or is involved in their production (last paragraph of the introduction). Please, clarify.
- The review shows different animal models for neurodegenerative diseases, but there is not much information through the manuscript about them. A summary table for the animal models presented in this work may improve this.
- I suggest to talk about the use of neurotrophic factors as a therapeutic approach in other neurodegenerative diseases, such as ALS, since many research has been done using NTSs in this disease.
- 4th line, 1st paragraph of the section “Rheb-mTORC1 signalling pathway as a potential therapeutic target for neurodegenerative diseases”: You say “Growth factors such as insulin and insulin-like growth factor 1….”. Please, notice insulin is a hormone.
- Which kind of approach is used to deliver the neurotrophic factors in the studies you mention in the 4th paragraph of the section “The importance of neurotrophic factor therapeutic strategy for neurodegenerative diseases”? Please, make it more clear in the manuscript.
- There is no needed to include the last sentence in the figure legend: “Overall, these observations……”
Author Response
Review 2
In this article, Nam and collaborators review the involvement of neurotrophic factors in certain neurodegenerative diseases and the importance of targeting them in order to treat those diseases. Moreover, they talk about the use of a modified version of the Rheb protein delivered by AAV1 vectors to target the mTOR pathway and to enhance the expression of neurotrophic factors.
The review is comprehensive and very well-organized. However, some major and minor points should be addressed before being accepted.
Major point:
- Comment: Some parts of the manuscript, such as “The importance of neurotrophic factor therapeutic strategy for neurodegenerative diseases” or “Rheb-mTORC1 signalling pathway as a potential therapeutic target for neurodegenerative diseases” should be re-written with a more appropriate and clear English style.
Response: We thank the reviewer for their comment. We have revised the text accordingly.
(Page 5 in revised manuscript) Importance of supporting neurotrophic factors as a therapeutic strategy for neurodegenerative diseases
(Page 13 in revised manuscript) Rheb-mTORC1 signaling against neurodegenerative diseases
Minor points:
- Comment: Please, carefully proof-read the entire manuscript to eliminate some grammatical errors and to avoid some word repetitions. It should be desirable to use more synonyms.
Response: We thank the reviewer for this comment. We have checked the manuscript and corrected the grammatical errors.
- Comment: It is not clear in the introduction whether Rheb can be activated by NTFs (line 6 of the second paragraph) or is involved in their production (last paragraph of the introduction). Please, clarify.
Response: We apologize for not clearly explaining the role of Rheb.
We have added an explanation to the introduction section of the revised manuscript (pages 4-5 in the revised manuscript).
Page 4 in the revised manuscript: The expression levels of Rheb are increased as an immediate early response to toxic stimulation, such as seizure and high-frequency-induced synaptic stimuli in an N-methyl-d-aspartate (NMDA)-dependent manner [26,27]. Further, Rheb can be activated by growth factors, such as nerve growth factor (NGF), epithelial growth factor, and fibroblast growth factor, in cultured neuronal cells [26].
Page 5 in the revised manuscript: Recently, it has been reported that activation of Rheb can protect neurons against neurotoxic conditions in the adult brain through neurotrophic interactions (such as BDNF, GDNF, and CNTF production) between neurons and astrocytes [39-42].
- Comment: The review shows different animal models for neurodegenerative diseases, but there is not much information through the manuscript about them. A summary table for the animal models presented in this work may improve this.
Response: As suggested by the reviewer, a list of all animal models presented in this study have been added to Table 2.
- Comment: I suggest to talk about the use of neurotrophic factors as a therapeutic approach in other neurodegenerative diseases, such as ALS, since many research has been done using NTSs in this disease.
Response: We thank the reviewer for this comment. As suggested by the reviewer, we have added text regarding the therapeutic approach and negative effects of neurotrophic factors in ALS.
Pages 10-11 in the revised manuscript: Various NTFs have been investigated in animal models for the treatment of other neurodegenerative diseases. ALS, the most common form of motor neuron disease, is a fatal adult-onset neurodegenerative disease that results in progressive and preferential degeneration and death of both the upper motor neurons of the motor cortex and the alpha lower motor neurons of the brain stem and spinal cord [111,112]. Several studies have documented the changes and neuroprotective effects of NTFs in ALS using in vivo and in vitro models. Expression levels of BDNF showed a significant reduction in spinal cord tissue obtained from SOD1 (G93A) mice, a murine model of ALS [62], and in lumbar spinal cord tissue of rat neonates that were injected intrathecally with CSF of ALS [63]. BDNF prevents cell death of motor neurons in the axotomized facial nucleus of the neonatal rat [113,114] and mediates antiapoptotic effects by the ERK and PI3K pathways [115]. Recently, studies have also confirmed that the modulation of TrkB via enhanced BDNF signaling increased neuronal survival in degenerating neurons in vitro [116] and improved motor dysfunction and motor neuron loss in ALS model mice [117]. GDNF has also been reported to have a protective effect on motor neurons in ALS. Disruption of the TNFR1-GDNF axis in astrocytes accelerates motor neuron degeneration and disease progression of ALS [118]. GDNF delivery prevents motor neurons from degenerating and preserves the axons that innervate the muscle; moreover, it has shown to inhibit muscle atrophy in a transgenic mice model with the G93A human SOD1 mutation of ALS [119]. Furthermore, four-limb injection of AAV-GDNF in ALS mice postpones disease onset, delays progression of motor dysfunction, and prolongs life span [120]. However, despite the observation of these neuroprotective effects, there is also evidence that shows that therapeutic intervention of BDNF is unable to promote survival or prevent neuronal death in vivo. Many studies have shown that BDNF negatively affects motor neuron survival, which makes motor neurons more susceptible to damage [121,122]. Moreover, BDNF is effective in enhancing excitotoxic damage by enhancing glutamatergic activity in neurons [123]. Several studies have reported that BDNF plays a key role in motor neuron susceptibility to excitotoxicity [122,124,125]. In addition, muscles and CSF of patients with ALS exhibit elevated levels of BDNF [126] and GDNF [127]. These results suggest that the negative effects of NTFs also need to be considered for NTF therapeutic strategies for ALS.
- Comment: 4th line, 1st paragraph of the section “Rheb-mTORC1 signaling pathway as a potential therapeutic target for neurodegenerative diseases”: You say “Growth factors such as insulin and insulin-like growth factor 1….”. Please, notice insulin is a hormone.
Response: We have modified the text accordingly.
Page 14 in the revised manuscript: Insulin and insulin-like growth factor 1 (IGF1), which activate G-protein-coupled and IGF1 receptors on target cell membranes, trigger the lipid kinase phosphatidylinositol-3 kinase (PI3K)-serine/threonine kinase Akt signaling pathway.
- Comment: Which kind of approach is used to deliver the neurotrophic factors in the studies you mention in the 4th paragraph of the section “The importance of neurotrophic factor therapeutic strategy for neurodegenerative diseases”? Please, make it more clear in the manuscript.
Response: We appreciate the reviewer’s comments. We have added text regarding the clinical trials of neurotrophic factor delivery using intracerebroventricular injection and intraputamenal injection in patients with PD.
Pages 9-10 in the revised manuscript: Trials investigating treatment of PD patients with GDNF have been in progress for 20 years. The initial research administered GDNF into the lateral ventricle of PD patients by the intracerebroventricular route [102]. In a multicenter, randomized, double-blind, placebo-controlled, sequential cohort study, 50 patients received GDNF monthly for 8 months and extended exposure up to an additional 20 months. However, no therapeutic effect was confirmed even at high concentrations of GDNF, and only side effects, such as nausea, weight loss, and asymptomatic hyponatremia, were reported. The lack of clinical effects in this trial was due to the possibility that GDNF was not effectively delivered into the target tissues, such as the putamen and SN, when administered intracerebroventricularly. This led to the consideration of other strategies for the administration of GDNF, and subsequent trials used intraparenchymal administration of GDNF. In an open-label study, five patients with PD were treated with intraputamenal infusion of GDNF [103]. Patients receiving GDNF showed a 30%–60% improvement in the off-medication motor sub-score of the Unified Parkinson’s Disease Rating Scale (UPDRS) and a 61% improvement in the activities of daily living sub-score. Moreover, the beneficial effect of GDNF administration was confirmed by increased putamenal 18F-dopa uptake measured by positron emission tomography (PET). In addition, it was confirmed that TH-immunopositive nerve fibers were increased in the postmortem brain tissue study of a patient who had received unilateral infusion for 48 months [104]. In other open-label studies, unilateral intraputamenal infusions of GDNF were performed in 10 patients with PD, and motor dysfunction was improved without any serious side effects for 6 and 12 months [105,106]. Intraputamenal infusion of GDNF has shown promise for GDNF as a factor for clinical improvement in PD patients, and based on these results, a new clinical trial was conducted. GDNF was continuously administered into the posterior dorsal putamen using a chronic infusion pump, and the patient was evaluated for 6 months [107]. After 6 months, there was no difference from the placebo group, and some patients had problems developing neutralizing antibodies to GDNF. The study was withdrawn because of such stability issues; however, one patient experienced clinical improvement several years following GDNF treatment [108]. In another study, GDNF was administered intraputamenally every month for 6 months using a convection-enhanced delivery system [109,110]. There was no significant improvement in UPDRS score in GDNF-administered patients, but an increase in dopamine neuron function was confirmed by PET imaging. As a new strategy for delivering GDNF into target tissue, a method using viral vectors is currently being considered. AAV2, which enables long-term expression of transgenes without inducing an inflammatory response, is currently the vector of choice for clinical trials in PD patients. Currently, AAV2-GDNF is administered to the putamen, and treatment studies are ongoing (NCT04167540 and NCT01621581).
- Comment: There is no needed to include the last sentence in the figure legend: “Overall, these observations……”
Response: As suggested by the reviewer, we have removed the last sentence in the figure legend.
Round 2
Reviewer 1 Report
The authors did a great job answering the comments. They have corrected the shortcomings, including the addition of an informative table and figure. The authors have also added thorough discussion on areas that were not discussed enough in the first version of the manuscript as requested.
Overall, I find the revised manuscript fit for publication without further modifications.
Reviewer 2 Report
The reviewed manuscript by Nam et al has been greatly improved. Under my point of view, major and minor points have been correctly addressed so I recommend accepting the review article.
As a suggestion, I would change the expression “mice with HD” by “HD-mice” in the second paragraph of the introduction.